# Clouds damp the radiative impacts of Polar sea ice loss

**Authors:** Ramdane Alkama[1*], Patrick C. Taylor[2*], Lorea Garcia-San Martin[1], Herve Douville[3], Gregory Duveiller[1], Giovanni Forzieri[1], Didier Swingedouw[4] and Alessandro Cescatti[1]

**Affiliation:**

[1] European Commission, Joint Research Centre, Via E. Fermi, 2749, I-21027 Ispra (VA), Italy

[2] NASA Langley Research Center, Hampton, Virginia

[3] Centre National de Recherches Meteorologiques, Meteo-France/CNRS, Toulouse, France

[4] EPOC, Universite Bordeaux 1, Allée Geoffroy Saint-Hilaire, Pessac 33615, France

**\*Correspondence to:** Ramdane Alkama (ram.alkama@hotmail.fr)

Patrick C. Taylor (patrick.c.taylor@nasa.gov)

**Abstract**

Clouds play an important role in the climate system: (1) cooling the Earth by reflecting incoming sunlight to space and (2) warming the Earth by reducing thermal energy loss to space. Cloud radiative effects are especially important in polar regions and have the potential to significantly alter the impact of sea ice decline on the surface radiation budget. Using CERES data and 32 CMIP5 climate models, we quantify the influence of polar clouds on the radiative impact of polar sea ice variability. Our results show that the cloud shortwave cooling effect strongly influences the impact of sea ice variability on the surface radiation budget and does so in a counter-intuitive manner over the polar seas: years with less sea ice and a larger net surface radiative flux show a more negative cloud radiative effect. Our results indicate that $66 \pm 2\%$ of this change in the net cloud radiative effect is due to the reduction in surface albedo and the remaining $34 \pm 1\%$ is due to an increase in cloud cover/optical thickness. The overall cloud radiative damping effect is $56 \pm 2\%$ over the Antarctic and $47 \pm 3\%$ over the Arctic. Thus, present-day cloud properties significantly reduce the net radiative impact of sea ice loss on the Arctic and Antarctic surface radiation budgets. As a result, climate models must accurately represent present-day polar cloud properties in order to capture the surface radiation budget impact of polar sea ice loss and thus the surface albedo feedback.

## 1. Introduction

Solar radiation is the primary energy source for the Earth system and provides the energy driving motions in the atmosphere and ocean, the energy behind water phase changes, and for the energy stored in fossil fuels. Only a fraction (Loeb et al., 2018) of the solar energy arriving to the top of the Earth atmosphere (shortwave radiation, SW) is absorbed at the surface. Some of it is reflected back to space by clouds and by the surface, while some is absorbed by the atmosphere. In parallel, the Earth's surface and atmosphere emit thermal energy back to space, called outgoing longwave (LW) radiation, resulting in a loss of energy (Fig. 1). The balance between these energy exchanges determines Earth's present and future climate. The change in this balance is particularly important over the Arctic where summer sea ice is retreating at an accelerated rate (Comiso et al., 2008), surface albedo is rapidly declining, and surface temperatures are rising at a rate double that of the global average (Cohen et al., 2014; Graversen et al., 2008), impacting sub-polar ecosystems (Cheung et al., 2009; Post et al., 2013) and possibly mid-latitude climate (Cohen et al., 2014; Cohen et al. 2019).

Clouds play an important role in modifying the radiative energy flows that determine Earth's climate. This is done both by increasing the amount of SW reflected back to space and by reducing the LW energy loss to space relative to clear skies (Fig. 1). These cloud effects on Earth's radiation budget can be gauged using the Cloud Radiative Effect (CRE), defined as the difference between the actual atmosphere and the same atmosphere without clouds (Charlock and Ramanathan, 1985). The different spectral components of this effect can be estimated from satellite observations: the global average SW cloud radiative effect (SWcre) is negative since clouds reflect incoming solar radiation back to space resulting in a cooling effect. On the other hand, the LW cloud radiative effect (LWcre) is positive since clouds reduce the outgoing LW radiation to space generating a warming effect (Harrison et al., 1990; Loeb et al., 2018; Ramanathan et al., 1989).

Cloud properties and their radiative effects exhibits significant uncertainty in the polar regions (e.g., Curry et al. 1996; Kay and Gettelman 2009; Boeke and Taylor 2016; Kato et al. 2018). For instance, climate models struggle to accurately simulate cloud cover, optical depth, and cloud phase (Cesana et al., 2012; Komurcu et al., 2014; Kay et al. 2016). An accurate representation of polar clouds is necessary because they strongly modulate radiative energy fluxes at the surface, in the atmosphere, and at the TOA influencing the evolution of the polar climate systems. In addition, polar cloud properties interact with other properties of the polar climate systems (e.g., sea ice) and influence how variability in these properties affects the surface energy budget (Qu and Hall 2006; Kay and L'Ecuyer 2013; Sledd and L'Ecuyer 2019). Morevoer, Loeb et al. (2019) documented severe limitations in the representation of surface albedo variations and their impact on the observed radiation budget variability in reanalysis products, motivating the evaluation of radiation budget variability over the polar seas in climate models. In this study, we use the Clouds and the Earth's Radiant Energy System (CERES) top-of-atmosphere (TOA) and surface (SFC) radiative flux datasets and 32 Coupled Model Intercomparison Project (CMIP5) climate models to estimate the relationship between the CRE and the surface radiation budget in polar regions to improve our understanding of how clouds modulate the surface radiation budget.

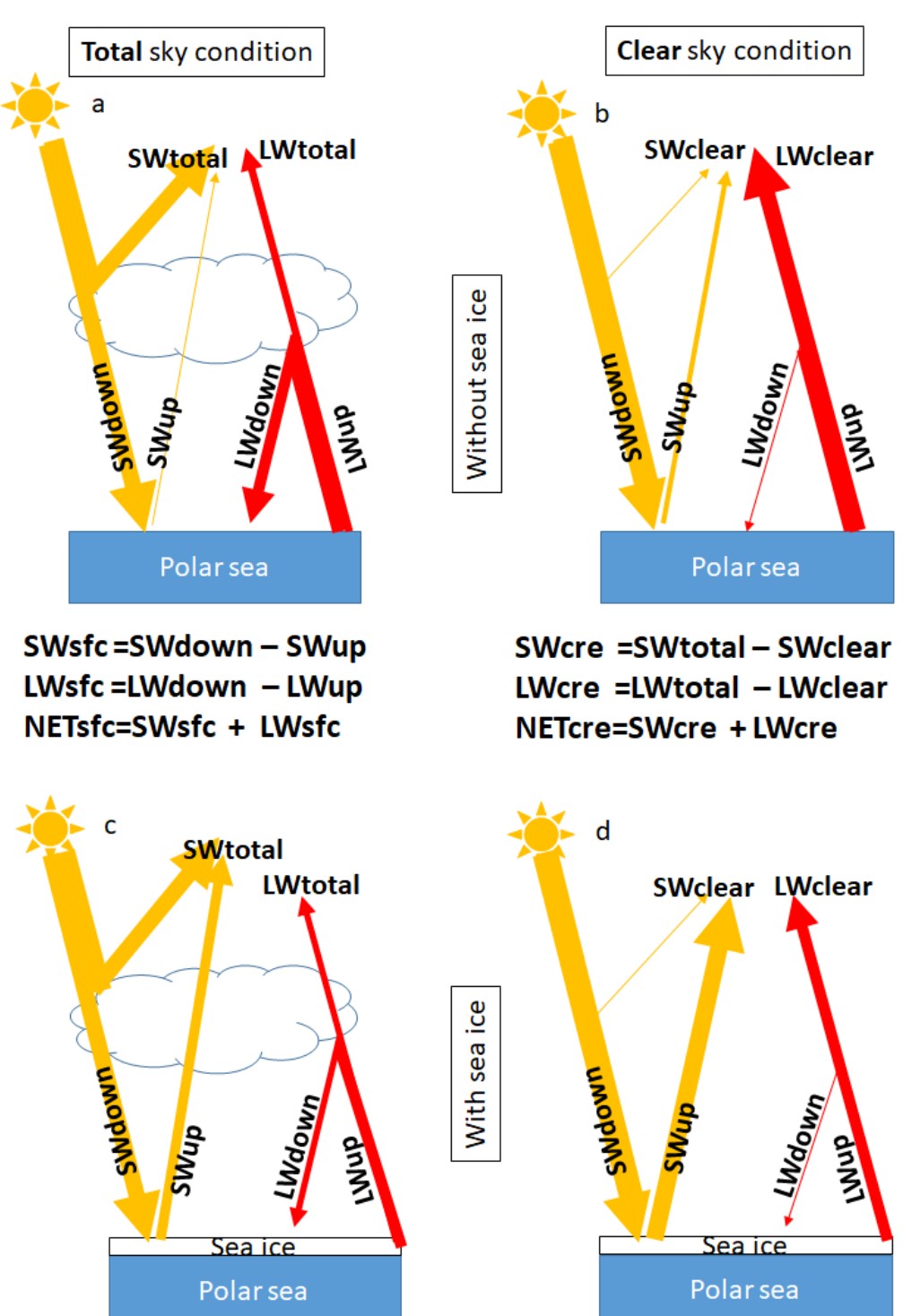

**Figure 1** Schematic representation of radiative energy flows in the polar seas under total sky conditions (a, c) and clear sky conditions (b, d) for two contrasting surface conditions: without sea ice (a, b) and with sea ice (c, d). All fluxes are taken positive downwards.

## 2. Methods and data

**2.1 CERES EBAF Ed4.0 Products:** Surface and TOA radiative flux quantities are taken from the NASA CERES Energy Balanced and Filled (EBAF) monthly data set (CERES EBAF-TOA_Ed4.0 and CERES EBAF-SFC_Ed4.0), providing monthly, global fluxes on a 1°x1° latitude-longitude grid (Loeb et al., 2018; Kato et al. 2018). CERES surface LW and SW radiative fluxes are used to investigate the effect of clouds on the surface radiation budget response to sea ice variability over the polar seas. CERES SFC EBAF radiative fluxes have been evaluated through comparisons with 46 buoys and 36 land sites across the globe, including the available high-quality sites in the Arctic. Uncertainty estimates for individual surface radiative flux terms in the polar regions range from 12-16 W m$^{-2}$ (1$\sigma$) at the monthly mean 1°x1° gridded scale (Kato et al. 2018). CERES EBAF-TOA and SFC radiative fluxes show a much higher reliability than other sources (e.g., meteorological reanalysis) and represent a key benchmark for evaluating the Arctic surface radiation budget (Christensen et al. 2016; Loeb et al. 2019; Duncan et al. 2020).

In addition to radiative fluxes, cloud cover fraction (CCF) and cloud optical depth (COD) data available from CERES EBAF data are used. Monthly mean CCF and COD data are derived from instantaneous cloud retrievals using the Moderate-resolution Imaging Spectroradiometer (MODIS) radiances (Trepte et al. 2019). Instantaneous retrievals are then are spatially and temporally averaged onto the 1°x1° monthly mean grid consistent with CERES EBAF.

**2.2 Cloud Radiative Effect:** CRE is used as a metric to assess the radiative impact of clouds on the climate system, defined as the difference in net irradiance at TOA between total-sky and clear-sky conditions. Using the CERES Energy Balanced And Filled (EBAF) Ed4.0 (Loeb et al., 2018) flux measurements and CMIP5 simulated fluxes, CRE is calculated by taking the difference between clear-sky and total-sky net irradiance flux at the TOA. All fluxes are taken as positive downwards.

$$SW_{cre}=SW_{total} - SW_{clear} \qquad (1)$$

$$LW_{cre}=LW_{total} - LW_{clear} \qquad (2)$$

$$NET_{cre}=SW_{cre} + LW_{cre} \qquad (3)$$

**2.3 Earth's surface radiative budget:** Surface radiative fluxes are taken from the CERES SFC EBAF Ed4.0 data set (Kato et al., 2018). The net SW and LW fluxes at the surface ($SW_{sfc}$ and $LW_{sfc}$, respectively) are calculated as the difference between the downwelling $SW_{down}$ ($LW_{down}$) and upwelling $SW_{up}$ ($LW_{up}$) as shown in equations 4 (5).

$$SW_{sfc}=SW_{down} - SW_{up} \qquad (4)$$

$$LW_{sfc}=LW_{down} - LW_{up} \qquad (5)$$

$NET_{sfc} = SW_{sfc} + LW_{sfc}$          (6)
**2.4 Sea ice concentration:** Sea ice concentration (SIC) data are from the National Snow and Ice
Data Center (NSIDC, http://nsidc.org/data/G02202). This data set is a Climate Data Record (CDR)
of SIC from passive microwave data and provides a consistent, daily and monthly time series of
SIC from 09 July 1987 through the most recent processing for both the North and South Polar
regions (Peng et al., 2013; W. Meier, F. Fetterer, M. Savoie, S. Mallory, R. Duerr, 2017). The data
is provided on a 25 km x 25 km grid. We used the latest version (Version 3) of the SIC CDR
created with a new version of the input product, from Nimbus-7 SMMR and DMSP SSM/I-SSMIS
Passive Microwave Data.
**2.5 Polar seas:** We define the polar seas as ocean regions where the monthly SIC is larger than
10% at least one month during 2001-2016 period. Polar seas extent is shown in Figure S1.
**2.6 CMIP5 Models** To reconstruct the historical CRE and surface energy budget and project their
future changes, we used an ensemble of simulations conducted with 32 climate models (models
used are shown in Figure 3 and S3) contributing to the Coupled Model Intercomparison Project
Phase 5 (CMIP5) (Taylor et al., 2012). These model experiments provided: historical runs (1850-
2005) in which all external forcings are consistent with observations and future runs (2006-2100)
using the RCP8.5 emission scenarios (Taylor et al., 2012). The comparison with the satellite data
is made over 2001-2016 by merging historical runs 2001-2005 with RCP8.5 2006-2016.
**2.7 Estimation of the local variations in radiative flux, cloud cover, and cloud optical depth**
**concurrent with changes in sea ice concentration**
This study employs a novel method for quantifying the variations in radiative fluxes and cloud
properties with SIC. This methodology leverages inter-annual variability of sea ice cover to assess
these relationships. Figure 2 schematically shows the methodology based on the following steps.
We use SW as an example and apply the approach in the same way to other variables.
1) $\Delta SW_j$ values are summarized in a schematized plot (Figure 2a) where each cell $j$ in such plot
shows the average $\Delta SW_m$ observed for all possible combinations of SIC at a grid box between two
consecutive observation years (year yi and yi+1 from time period 2001-2016) displayed on the X
and Y axes, respectively. For the sake of clarity in Figure 2 the X and Y axes report SIC in intervals
of 10%, while in Figure 5, 6, 7, S5 and S6 the axes are discretized with 2% bins.
2) Because of the regular latitude/longitude grid used in the analysis, the area of the grid cells ($a_m$)
varies with the latitude. The energy signal ($\Delta SW_j$) is therefore computed as an area weighted
average (Equation 7) where M is the number of grid cells that are used to compute cell $j$ in the
schematised plot Fig 2a. Figure 2b shows the total area of all these grid cells as described by
Equation 8.
$$\Delta SW_j = \frac{\sum_{m=1}^{M} a_m \Delta SW_m}{\sum_{m=1}^{M} a_m}$$          (7)

153 $A_j = \sum_{m=1}^{M} a_m$       (8)


155 3) Calculation of the area weighted average ($\Delta SW_p$) of the energy signal of all $N$ cells with the
156 same fraction $X$ of a change in SIC (shown with the same colour in Figure 2a) Equation 9.

157 $\Delta SW_p = \dfrac{\sum_{j=1}^{N} A_j \Delta SW_j}{\sum_{j=1}^{N} A_j}$       (9)

158 $\sum_{j=1}^{N} A_j$ is the total area of all grid cells with a particular SIC change.

159 $\Delta SW_p$ is the energy weighted average of all grid cells with a particular SIC change.

160

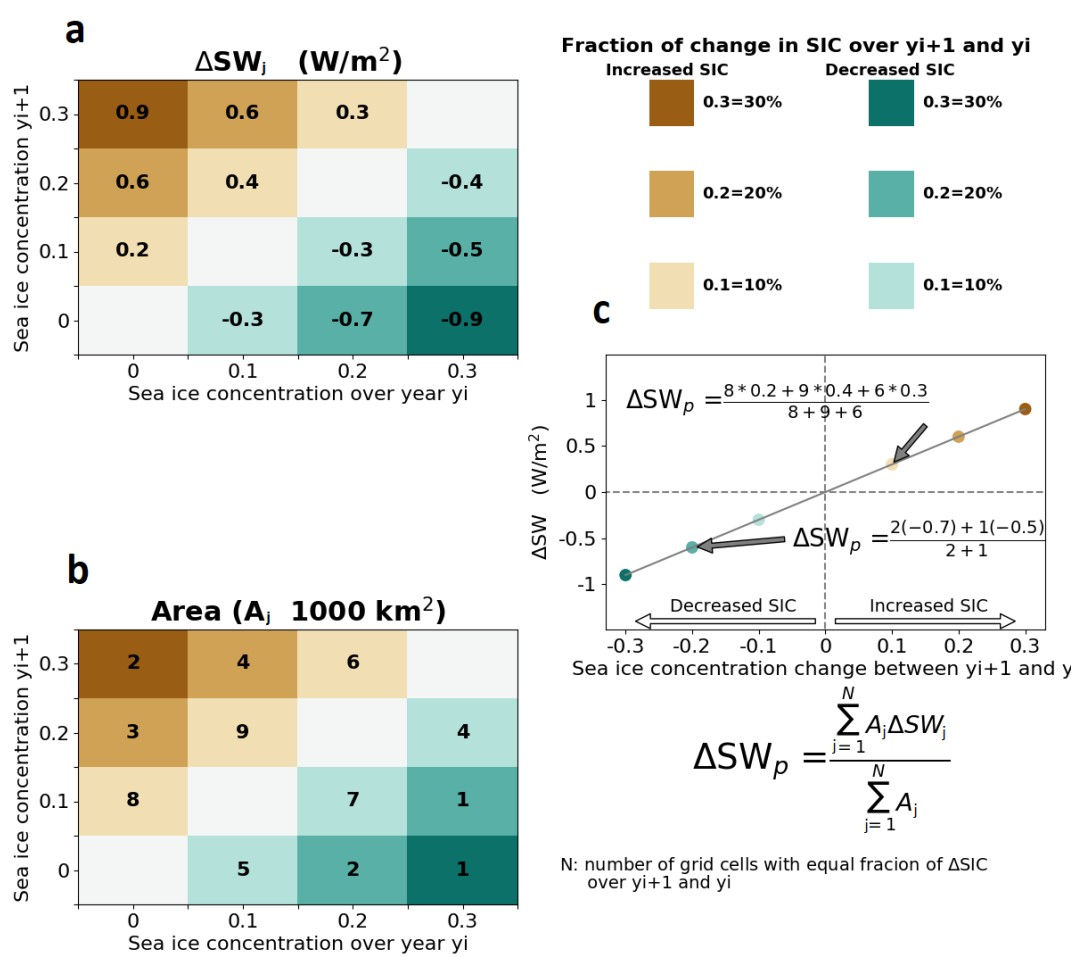

161

162 **Figure 2** Schematic representation of the methodology used to quantify the energy flux sensitivity
163 to changes in sea ice concentration as a linear regression between the percentage of sea ice
164 concentration and the variation in energy flux (right panel) using SW energy flux data and sea ice
165 concentration defined in the left panels.


The average energy signals $(\Delta SW_p)$ per class of sea ice concentration change are reported in a
scatterplot (Fig. 2 right panel) and used to estimate a regression line with zero intercept.
The slope $S$ of this linear regression represents the local SW energy signal generated by the
complete sea ice melting of a 1° grid cell. The weighted root mean square error (WRMSE) of the
slope is estimated by Equation 10, where $p$ represents one of the $NP$ points in the scatterplot (Fig.
2 right panel) and $X_p$ is the relative change in sea ice concentration in the range ±1 (equivalent to
±100% of sea ice cover change).
$$WRMSE = \sqrt{\frac{\sum_{p=1}^{NP} A_p (\Delta SW_p - S X_p)^2}{\sum_{p=1}^{NP} A_p}}, \quad \text{where} \quad A_p = \sum_{j=1}^{N} A_j \qquad (10)$$
**2.8 Diagnosis of contributions to SWcre**
SWcre at the surface for the year yi (Eq. 11) and year yi+1 (Eq. 12) is function of surface albedo
$\alpha$, SWdown under clear sky conditions $(SW \downarrow_{clr})$ and SWdown under total sky conditions
$(SW \downarrow_{tot})$.
$$SWcre_{yi} = (1 - \alpha_{yi})(SW \downarrow_{tot,yi} - SW \downarrow_{clr,yi}) \qquad (11)$$
$$SWcre_{yi+1} = (1 - \alpha_{yi+1})(SW \downarrow_{tot,yi+1} - SW \downarrow_{clr,yi+1}) \qquad (12)$$

Using the first-order Taylor series expansion to (11) yields
$$\Delta SWcre_{yi+1-yi} \cong$$
$$(-\Delta\alpha_{yi+1-yi})(SW \downarrow_{tot,yi} - SW \downarrow_{clr,yi}) + (1 - \alpha_{yi})\Delta_{yi+1-yi}(SW \downarrow_{tot} - SW \downarrow_{clr}) \qquad (13)$$

Where
$$\Delta_{yi+1-yi}(SW \downarrow_{tot} - SW \downarrow_{clr}) \cong (SW \downarrow_{tot,yi+1} - SW \downarrow_{clr,yi+1}) - (SW \downarrow_{tot,yi} - SW \downarrow_{clr,yi}) \quad (14)$$

Separating the terms yields,
$$\Delta SWcre_{Alb} \cong (-\Delta\alpha_{yi+1-yi})(SW \downarrow_{tot,yi} - SW \downarrow_{clr,yi}) \quad (15)$$
Where $\Delta SWcre_{Alb}$ is the part of SWcre change that is induced by the change in surface albedo.

$$\Delta SWcre_{cloud} \cong (1 - \alpha_{yi})\Delta_{yi+1-yi}(SW \downarrow_{tot} - SW \downarrow_{clr}) \qquad (16)$$
Where $\Delta SWcre_{cloud}$ is the part of SWcre change that is induced by the change in cloud cover and cloud
optical depth.
$\Delta SWcre_{yi+1-yi} \cong \Delta SWcre_{Alb} + \Delta SWcre_{cloud}$ (17).
The above equations are used in figure 7 and S5.

**3. Results and discussions**
**3.1 Negative correlation patterns between cloud radiative effect and surface radiation on**
**polar seas**

Given the known cloud influence on the surface radiative budget, a positive correlation between
TOA CRE and surface radiative budget is expected (the amount of absorbed radiation at the surface
decreases with a more negative SWcre and a less positive LWcre). Figure 3 illustrates a positive
correlation between the annual mean NETcre and NETsfc over much of the global ocean using the
CERES TOA flux data from 2001-2016. However, our analysis reveals the opposite pattern over
the polar seas (defined in section 2.5) where the correlation is negative over the Antarctic and
partly negative over the Arctic (Bering Strait, Hudson Bay, Barents Sea and the Canadian
Archipelago; Fig. 3ab). Considering the SWcre and LWcre components, we find that the SWcre
(Fig. 3cd) shows a similar pattern of correlation as the NETcre (Fig. 3ab) but with a stronger
magnitude, while LWcre generally shows the opposite correlations (Fig. 3ef). This suggests that
the factors influencing SWcre are responsible for the sharp contrast in the correlation found in the
polar regions. Indeed, SWsfc and SWcre (Fig. 3gh) show the sharpest and most significant contrast
between the polar regions and the rest of the world (Fig. S2 is similar to Fig. 3 but only significant
correlations at the 95% confidence level are reported in blue and red colors). Overall, climate
models are able to reproduce the spatial pattern of the observed SW correlation, but also show a
large inter-model spread in the spatial extent of the phenomena (Fig. 4 and S3). On the other hand,
several models completely fail to reproduce the correlation. ACCESS1-3, MIROC5, CanESM2
and CSIRO-Mk3-6-0 models show negative correlation over Antarctic continent in contrast to
observed positive correlation. Some models, like IPSL-CM5B-LR, GISS-E2-R and bcc-csm1-1,
fail to reproduce the observed negative correlation over the Southern Ocean. This suggests that
these models contain misrepresentations of the relationships SWcre and NETsfc likely resulting
from errors in the relationships between sea ice, surface albedo, cloud cover/thickness, and their
influence on surface radiative fluxes that could severely impact their projections. Moreover, Fig.
4 demonstrates that simple correlations between NETsfc and the individual radiation budget terms
represents a powerful metric for climate model evaluation allows for a quick check for realistic
surface radiation budget variability in polar regions.

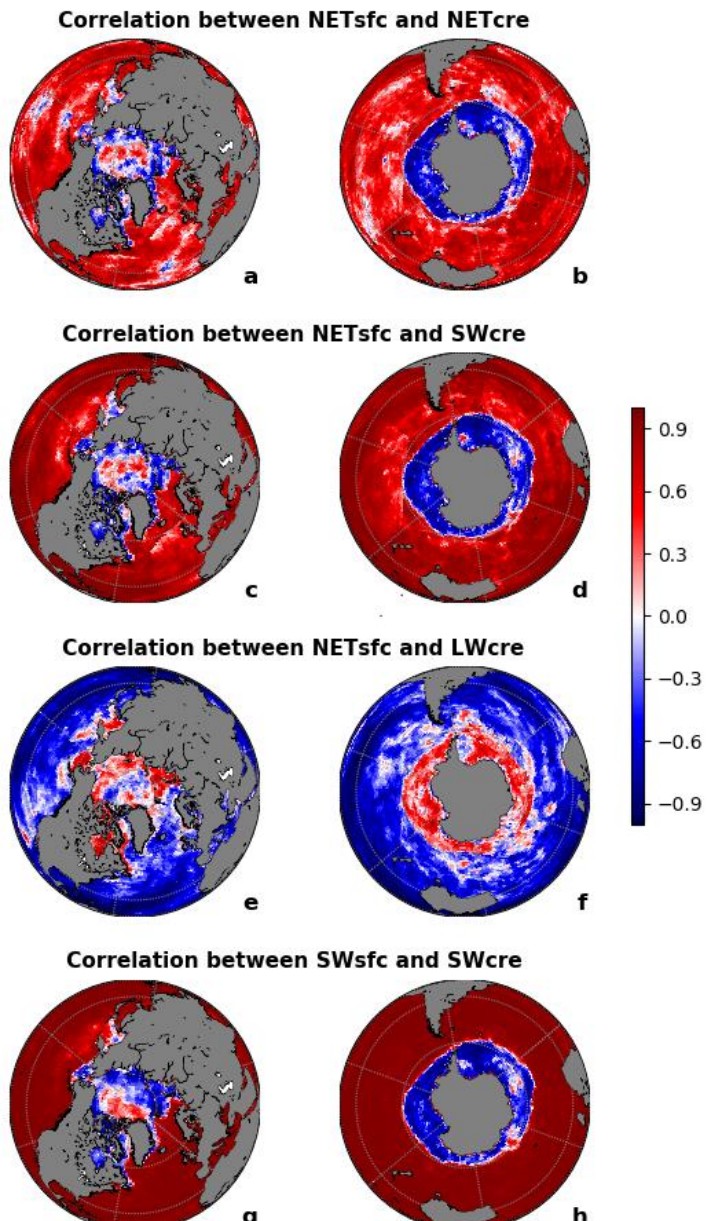


**Figure 3** Correlation between TOA CRE and surface radiation budget terms over 2001-2016 from CERES measurements for the Northern Hemisphere (aceg) and Southern Hemisphere (bdfh) polar sea. Positive correlations shown by the red color indicate that years with less NETsfc coincide with years where NETcre has a stronger cooling effect and *vice versa*.

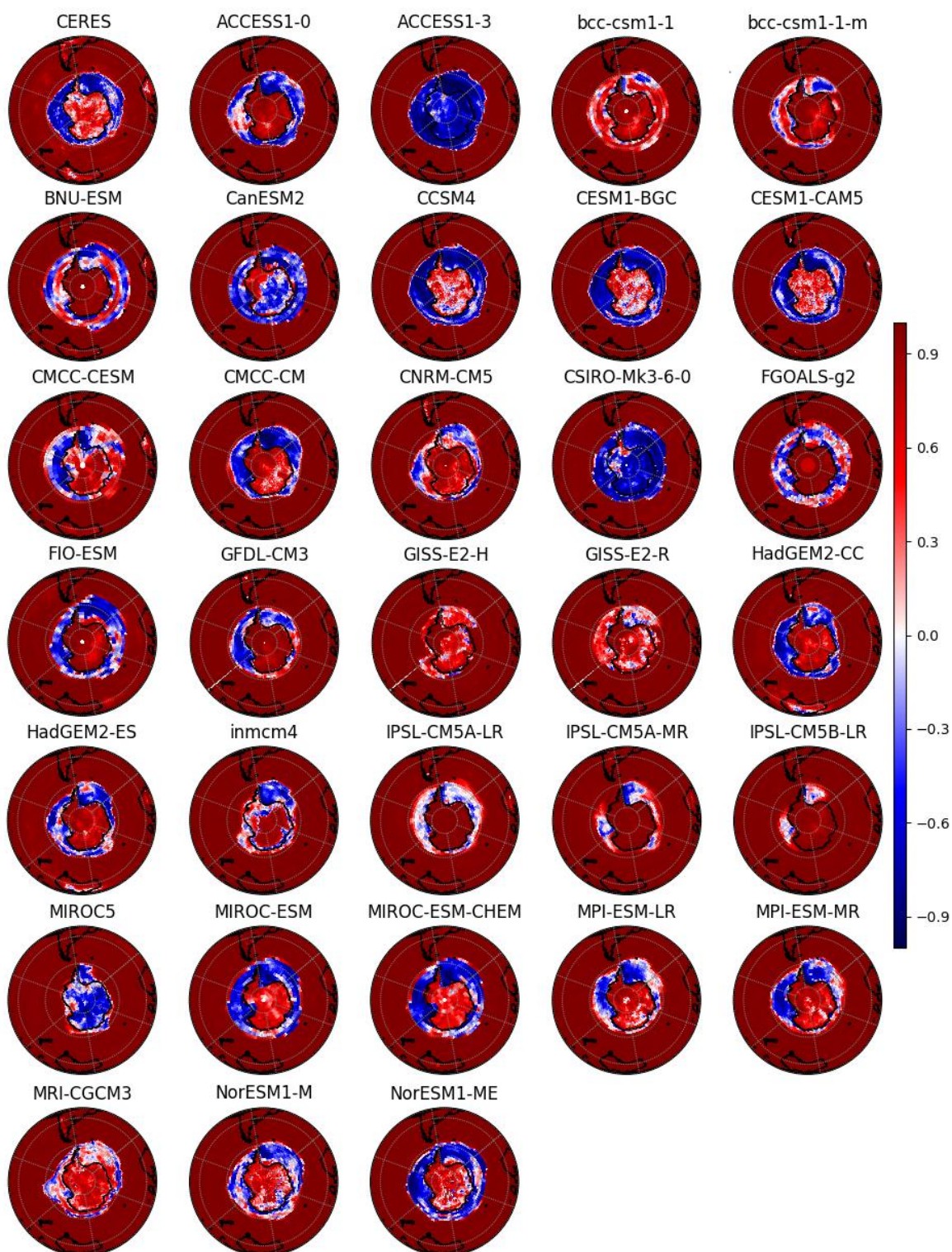

Figure 4 Correlation between SWcre and SWsfc shown by 32 CMIP5 earth system models and CERES between 2001 and 2016 over the Southern Hemisphere.

## 3.2 Effects of sea ice concentration change

We illustrate that the apparent contradiction over the polar seas between NETcre and NETsfc found in Fig 3ab is caused by the factors contributing to the SW fluxes. This can be explained by: (I) SWcre can change even if cloud properties are held constant due to the changes in clear-sky radiation induced by changes in sea ice and surface albedo. When surface albedo is reduced, the surface absorbs more sunlight at the surface resulting in a greater SWtotal. At the same time, SWclear increases since the lower albedo allows a larger fraction of the extra downwelling SW at the surface to be absorbed (see Fig. 1). Therefore, SWcre becomes more negative even in the absence of cloud changes (a purely surface-related effect); (II) On the other hand, the relationship between cloud cover/thickness and sea ice could lead to cloudier Polar seas under melting sea ice (Abe et al., 2016; Liu et al., 2012) such that the SWcre decreases (increasing the amount of SW reflected back to space by clouds, see Fig. 1), thus the cloud cooling effect is enhanced concurrently with melting sea ice (a purely cloud-related effect). Both of these factors occur simultaneously.

Over the Antarctic Ocean, analysis of the year-to-year changes in SWdown stratified in 2% SIC bins retrieved from satellite microwave radiometer measurements (see section 2.7) shows an increase in SWdown with increased SIC and *vice-versa* (Fig. 5a). This suggests that years with higher SIC also have fewer and/or thinner clouds (Liu et al., 2012) (Fig. 6), larger SWdown and also larger upward SW radiation (SWup) (Fig. 5b), due to higher surface albedo (Fig. S4). Consequently, these years show a more negative SWsfc (Fig. 5c) and thus are characterized by stronger surface cooling. Furthermore, fewer clouds implies a reduction of the cloud cooling effect (less negative SWcre) as described above in process (II), this accounts for $(19.42 * 100)/56.59 = 34 \pm 1\%$ (Fig. 7d bottom) of the total change in SWcre, and as described in process (I) the increase in the surface albedo also makes SWcre less negative and explains $(37.17 * 100)/56.59 = 66 \pm 2\%$ of the observed change (Fig. 7d bottom). Thus, the observed negative correlation between SWcre and SWsfc over the polar seas results from the larger effects of process (I) than (II). Similar results are found over the Arctic Ocean with slightly different sensitivity (Fig. S5, S6). This difference is tied to differences in sun angle/available sunlight, as Antarctic sea ice is concentrated at lower latitudes than Arctic sea ice.

Using the regression relationships derived from our composite analysis, we estimate the magnitude of the cloud effect. For the Antarctic system, we use the numbers found in Figure 5e where we find the annual mean relationship between NETsfc (in W/m$^2$) and SIC (fraction between 0 and 1), and NETcre (in W/m$^2$) and SIC (fraction between 0 and 1).

$\Delta$NETsfc=(-36.61$\pm$0.72)$\Delta$SIC  (18)

$\Delta$NETcre=(47.03$\pm$1.01)$\Delta$SIC  (19)

When excluding the CRE, the $\Delta$NETsfc would be equal to (-36.61-47.03) $\Delta$SIC =-83.64 $\Delta$SIC.

We estimate that the existence of clouds and their property variations are damping the potential

increase in the NETsfc within the Antarctic system due to the surface albedo decrease from sea ice
melt by 56% (47.03/83.64). The uncertainty is calculated by summing the uncertainties shown in
equation (18) and (19) as follows: $(0.72^2+1.01^2)^{1/2}/83.64=2\%$.
Similarly, over the Arctic (Fig. S5), we compute the cloud influence on the surface net radiative
budget that covaries with sea ice loss is $47\pm3\%$, in agreement with the study of Sledd and L'Ecuyer
284    (2019).


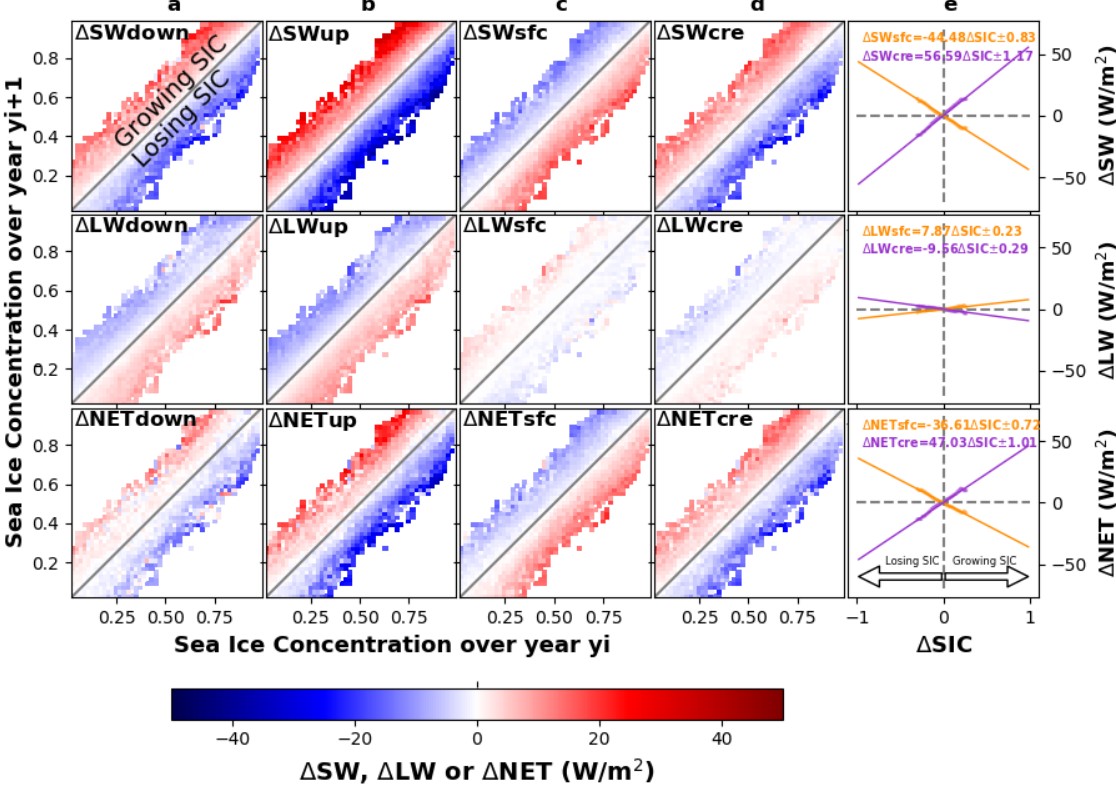

**Figure 5** Annual changes in SW, LW and NET as function of SIC. Annual changes in SW (top),
LW (middle) and NET (bottom) of radiative down (a), up (b), sfc=down-up (c) and cre (d) over
Antarctic Ocean as function of SIC change between two consecutive years $y_{i+1}$ and $y_i$ from 2001-
2016 time period. The top triangles in (c top) refers to the increase (growing) in SIC while the blue
color means a reduction (cooling) in SWsfc. Whereas, the top triangles in (d) refers to the increase
in SIC while the red color means an increase (decreasing the cooling role of clouds) in SWcre.
Each dot in column (e) represents the average of one parallel to the diagonal in (c) or (d) as
described in the Section 2.7.

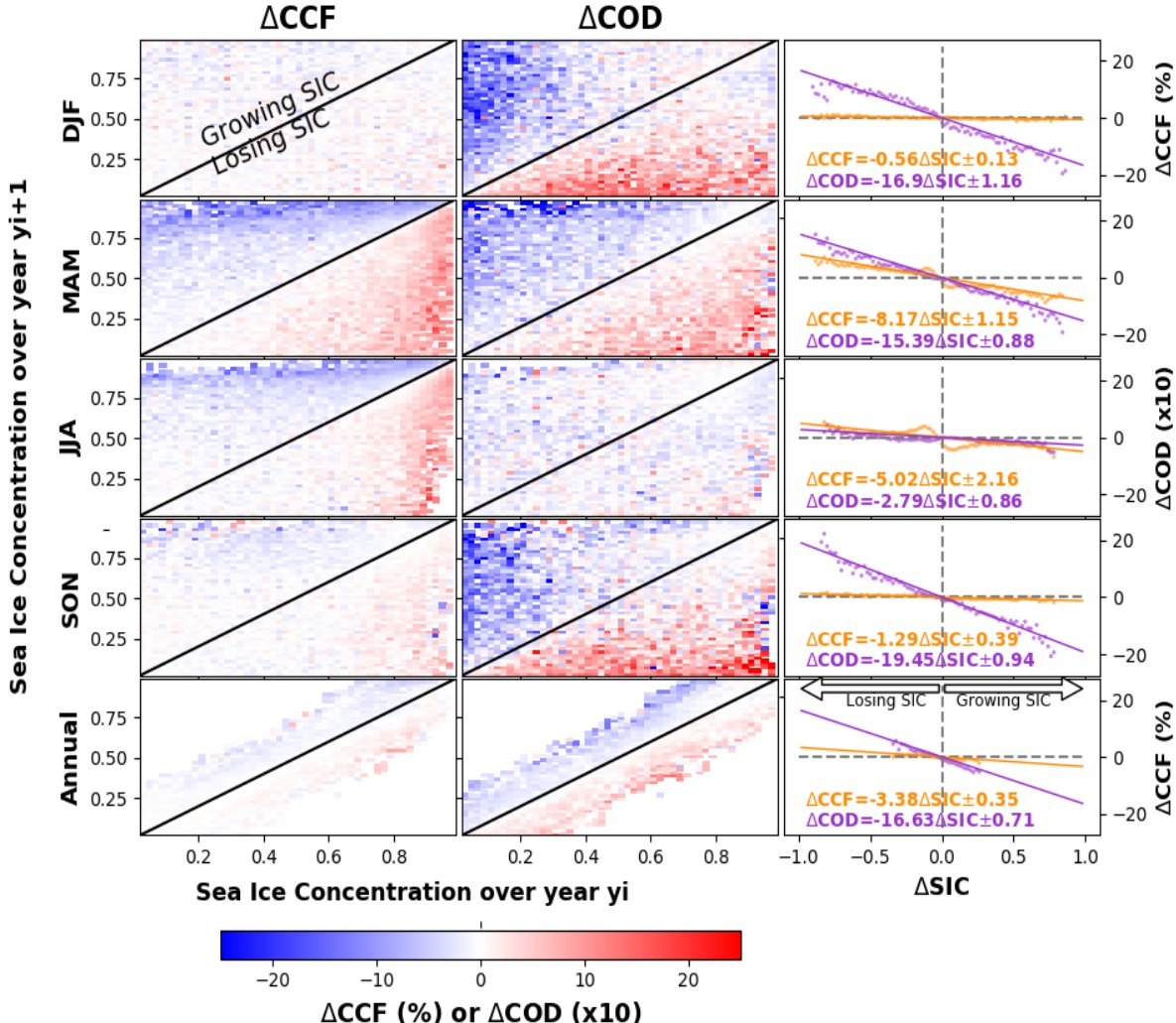

**Figure 6** Seasonal and annual changes in cloud cover fraction (CCF) and cloud optical depth (COD) over the Antarctic polar sea region as a function of SIC change between two consecutive years $y_{i+1}$ and $y_i$ from 2001-2016 time period. In order to use the same scale, COD has been multiplied by a factor 10. The top triangles in the two first columns refer to the increase (growing) in SIC while the blue color means a reduction in CCF or COD.

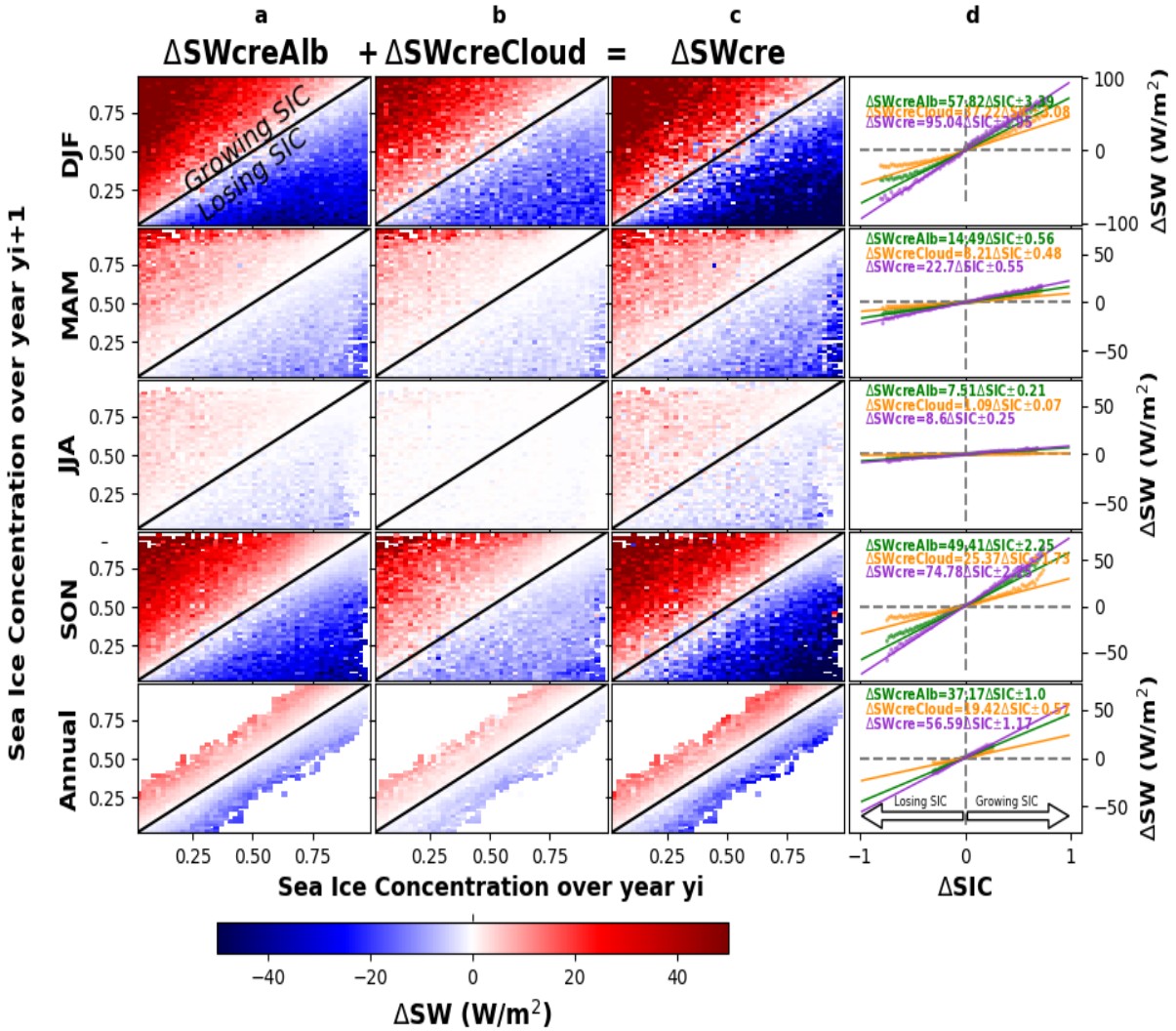

**Figure 7** Seasonal and annual changes in SWcreAlb, SWcreCloud and SWcre over the Antarctic polar sea region as function of SIC change between two consecutive years $y_{i+1}$ and $y_i$ from 2001-2016 time period. The analysis is based on method described in section 2.7 and observations from satellites data.

Altogether the results suggest clouds substantially reduce the impact of sea ice loss on the surface radiation budget and thus the observed sea ice albedo feedback. This effect in the polar climate system leads to a substantial reduction (56±2% over the Antarctic and 47±3% over the Arctic) of the potential increase in NETsfc in response to sea ice loss. This magnitude is similar to a previous study (Qu and Hall 2006) showing across a climate model ensemble that clouds damped the TOA effect of land surface albedo variations by half. Sledd and L'Ecuyer (2019) also determined that the cloud damping effect (also referred to as cloud masking) of the TOA albedo variability results from Arctic sea ice changes was approximately half. Despite this mechanism, the sharp reduction in Arctic surface albedo has dominated the recent change in the surface radiative budget and has led to a significant increase in NETsfc since 2001 in the CERES data (Duncan et al. 2020). These results demonstrate that the trends in polar surface radiative fluxes are driven by reductions in SIC and surface albedo and that clouds have partly mitigated the trend (i.e., a damping effect). Our findings highlight the importance of processes that control sea ice albedo (i.e. sea ice dynamics, snowfall, melt pond formation, and the deposition of black carbon), as the surface albedo of the polar seas in regions of seasonal sea ice is crucial for the climate dynamics.

### 3.3 Sensitivity of the surface energy budget to variability of sea ice concentration

Our results are consistent with other recent studies (Taylor et al., 2015; Morrison et al. 2018) that demonstrate a CCF response to reduced sea ice in fall/winter but not in summer (Figure 8a) over the Arctic Ocean. The lack of a summer cloud response to sea ice loss is explained by the prevailing air-sea temperature gradient, where near surface air temperatures are frequently warmer than the surface temperature (Kay and Gettelman 2009). Surface temperatures in regions of sea ice melt hover near freezing due to the phase change, whereas the atmospheric temperatures are not constrained by the freezing/melting point. Despite reduced sea ice cover, increases in surface evaporation (latent heat) are limited (Fig. 8mn), as also suggested by the small trends in surface evaporation rate derived from satellite-based estimates (Boisvert and Stroeve, 2015; Taylor et al., 2018). We argue that the strong increase of SWcreCloud under decreased sea ice observed during summer is induced by larger values of COD (Fig. 8a), which depend on the liquid or ice water content. We also show that the relationships derived from our observation-driven analysis match the projected changes in the Arctic and Antarctic surface energy budget in the median CMIP5 model ensemble (Fig. 8). However, we find a large spread amongst climate models that indicates considerable uncertainty.

Analyzing the seasonal cycle of the sensitivity of the surface energy budget to SIC variability, we found that SWsfc (SWcre) explains most of the observed changes in the NETsfc (NETcre) during summer, while LWsfc plays a minor role (Fig. 8). In contrast, during winter LWsfc (LWcre) explains most of the observed changes in the NETsfc (NETcre). In general, the median of the 32 CMIP5 (Taylor et al., 2012) climate models captures the observed sensitivity of the radiative energy budget and cloud cover change to SIC but the spread between climate models is large, especially for CCF. We have to note here that, the numbers reported in Figure 8 are for 100% SIC loss, while the ones reported in the previous figures (Fig. 5, 6 and 7) are for 100% SIC gain, explaining the opposite sign.

356

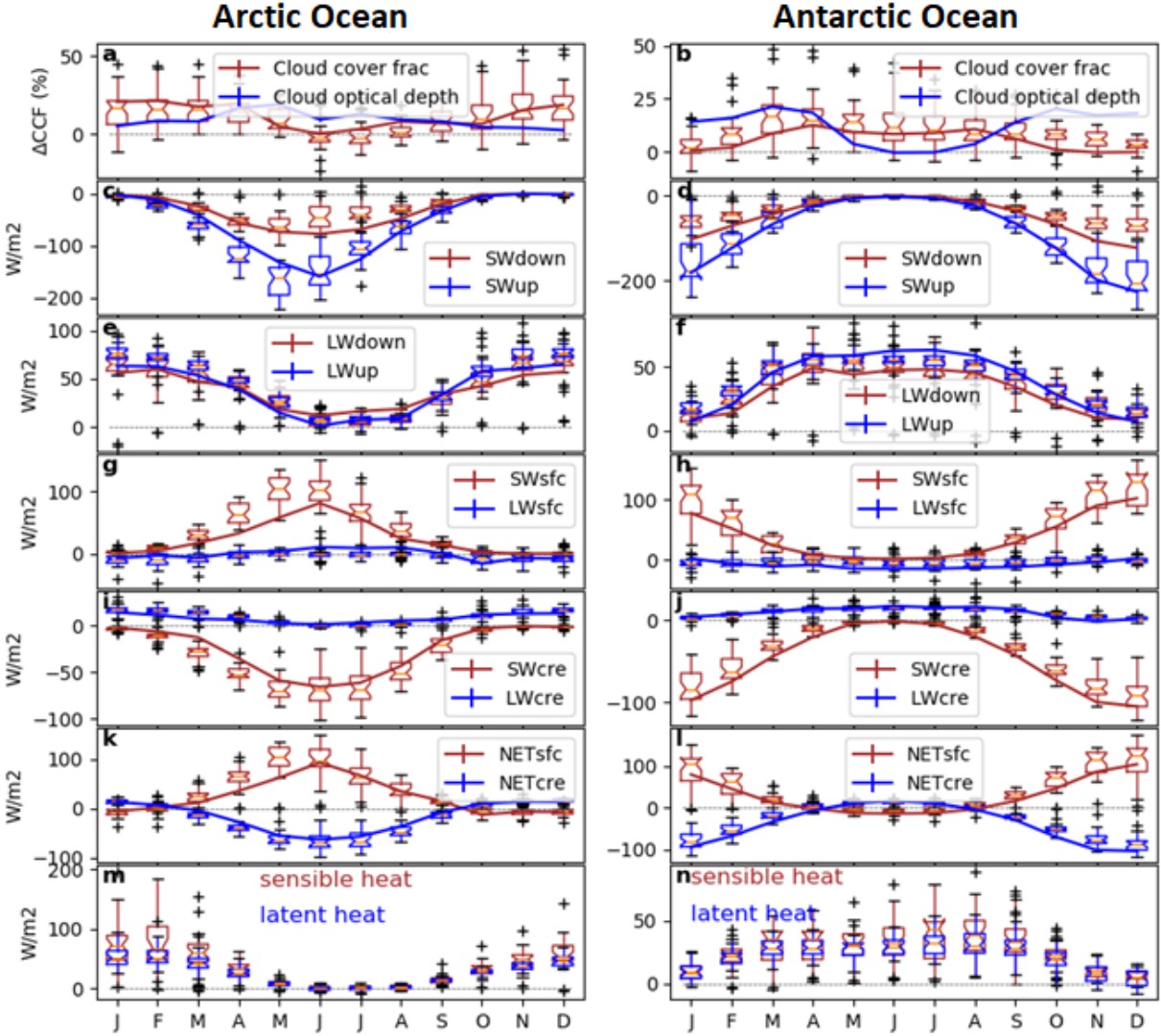

357
**Figure 8** Monthly change in different terms of the radiative energy balance, cloud optical depth
(COD) and cloud cover fraction (CCF) extrapolated from observations for a hypothetical 100%
decrease in SIC over the areas where SIC change was observed during the period 2001-2016. This
estimate came from the use of a linear interpolation of the change of different parts of the energy
budget, COD and CCF as function of a change in SIC coming from all possible combinations of
couplets of consecutive years for a given month from 2001 to 2016 and for all grid cells for which
SIC is larger than zero in one of the two years (see section 2.7). CERES data are shown by solid
lines (the standard deviation of the slopes are also reported but are too small to be visible) while
CMIP5 models are shown by boxplot and the box (are in same color as observations) represents
the first and third quartiles (whiskers indicate the 99% confidence interval and black markers show
outliers). In order to use the same scale, COD has been multiplied by a factor 10.

369
370

**3.4 Projections and uncertainties of cloud radiative effects on surface energy budget**

Under the RCP8.5 scenario ("business as usual"; Taylor et al., 2012), CMIP5 models show an increase in SWsfc over the Arctic Ocean (Fig. 9a), consistent with the expected decrease in the SIC (Comiso et al., 2008; Serreze et al., 2007; Stroeve et al., 2007). This increase in SWsfc occurs despite the large, concurrent and opposing change in SWcre. Projections of LW flux changes (Fig. 9c) are expected to play a small, but non-negligible role on total energy budget in summer by slightly increasing NETsfc (Fig. 9e). In addition, CMIP5 models indicate that by 2100 the magnitude of the NETcre decrease will be slightly smaller than the increase in NETsfc (Fig. 9e) over the Arctic Ocean; while, the Antarctic polar sea region shows the opposite (Fig. 9f). This is in line with the estimated damping effect of clouds coming from CERES over 2001-2016 that is about 47±3% in the Arctic and 56±2% in the Antarctic. The stronger cloud damping effect in the Antarctic region is indicated by the stronger negative change in NETcre in the Antarctic compared to the Arctic (Fig. 9ef).

Large uncertainties remain in the rate of summer sea ice decline and the timing of the first sea ice-free Arctic summer (Arzel et al., 2006; Zhang and Walsh, 2006). The processes responsible for the large inter-model spread between climate models are still under scrutiny (Holland et al., 2017; Simmonds, 2015; Turner et al., 2013). However, recent studies reaffirm the important role of the sea ice albedo feedback and the associated increased upper Arctic Ocean heat content (Holland and Lundrum 2015; Boeke and Taylor 2018) as well as the contributions from temperature-related feedbacks (Pithan and Maruitsen 2014; Stuecker et al. 2018). Figure 9gh shows that the annual mean Arctic and Antarctic sea ice extent trend from 32 CMIP5 models possesses a large positive correlation with the simulated trend in the SWdown, in line with previous studies (Holland and Lundrum 2015). We note that from the 32 CMIP5 models tested, only a few show SWdown trends consistent with observed trends in SWdown and SIC over 2001-2016 (Figure 9gh). Understanding the factors responsible for this disagreement between model-simulated and observed trends in SWdown and SIC may be provide insights into the processes responsible for the inter-model spread in Arctic climate change projections and are the subject of future work. We also find that the models with a larger trend in cloud cover also possess a larger decrease in sea ice extent, suggesting a stronger coupling between these two variables that may become stronger in the future. However, the direction of causality between the two variables is unclear and also requires further study.

**4. Conclusion**

The manuscript addresses two important climate science topics, namely the role of clouds and the fate of polar sea ice. The work is grounded in a long time series of robust satellite observations that allowed us to document an important damping effect in the polar cloud-sea ice system using a unique inter-annual approach. Our results agree with several previous works that approached the problem from a different perspective (Hartmann and Ceppi 2014; Sledd and L'Ecuyer 2019). In

addition, we show how 32 state-of-the-art climate models represent aspects of the surface radiation
budget over the polar seas.

Our data-driven analysis shows that polar sea ice and clouds interplay in a way that substantially
reduces the impact of the sea ice loss on the surface radiation budget. We found that when sea ice
cover is reduced between two consecutive years, the cloud radiative effect becomes more negative,
damping the total change in the net surface energy budget. The magnitude of this effect is
important. Satellite data indicates that the more negative cloud radiative effect reduces the
potential increase of net radiation at the surface by approximately half. One-third of this cloud
radiative effect change is induced by the direct change in cloud cover/thickness, while two-thirds
of this change results from the surface albedo change.

In addition, we demonstrated that the models that show larger trends in polar sea ice extent also
show larger trends in surface net solar radiation. In order to understand current and future climate
trajectories, model developments should aim at reducing uncertainties in the representation of
polar cloud processes in order to improve the simulation of present-day cloud properties over the
polar seas. Present-day Arctic and Antarctic cloud properties strongly influence the model
simulated cloud damping effect on the radiative impacts of sea ice loss.

Future cloud changes and sea ice evolution represent major uncertainties in climate projections
due to the multiple relevant pathways through which cloudiness and sea ice feed back on Earth's
climate system (Solomon et al. 2007). Our evidence derived from Earth observations provides
additional insight into the coupled radiative impacts of polar clouds and the changing sea ice cover
(Fig. 8) that may provide a useful constraint on model projections and ultimately improve our
understanding of present and future polar climate. On a practical level, our results demonstrate a
simple correlation analysis between the net surface radiation budget and individual radiation
budget terms that can be used to quickly evaluate climate models for realistic surface radiation
budget variability in polar regions. Ultimately, our findings on the interplay between cloud and
sea ice may support an improvement in the model representation of the cloud-ice interactions,
mechanisms that may substantially affect the speed of the polar sea ice retreat, which in turn has a
broad impact on the climate system, on the Arctic environment and on potential economic
activities in the Arctic region (Buixadé Farré et al., 2014).

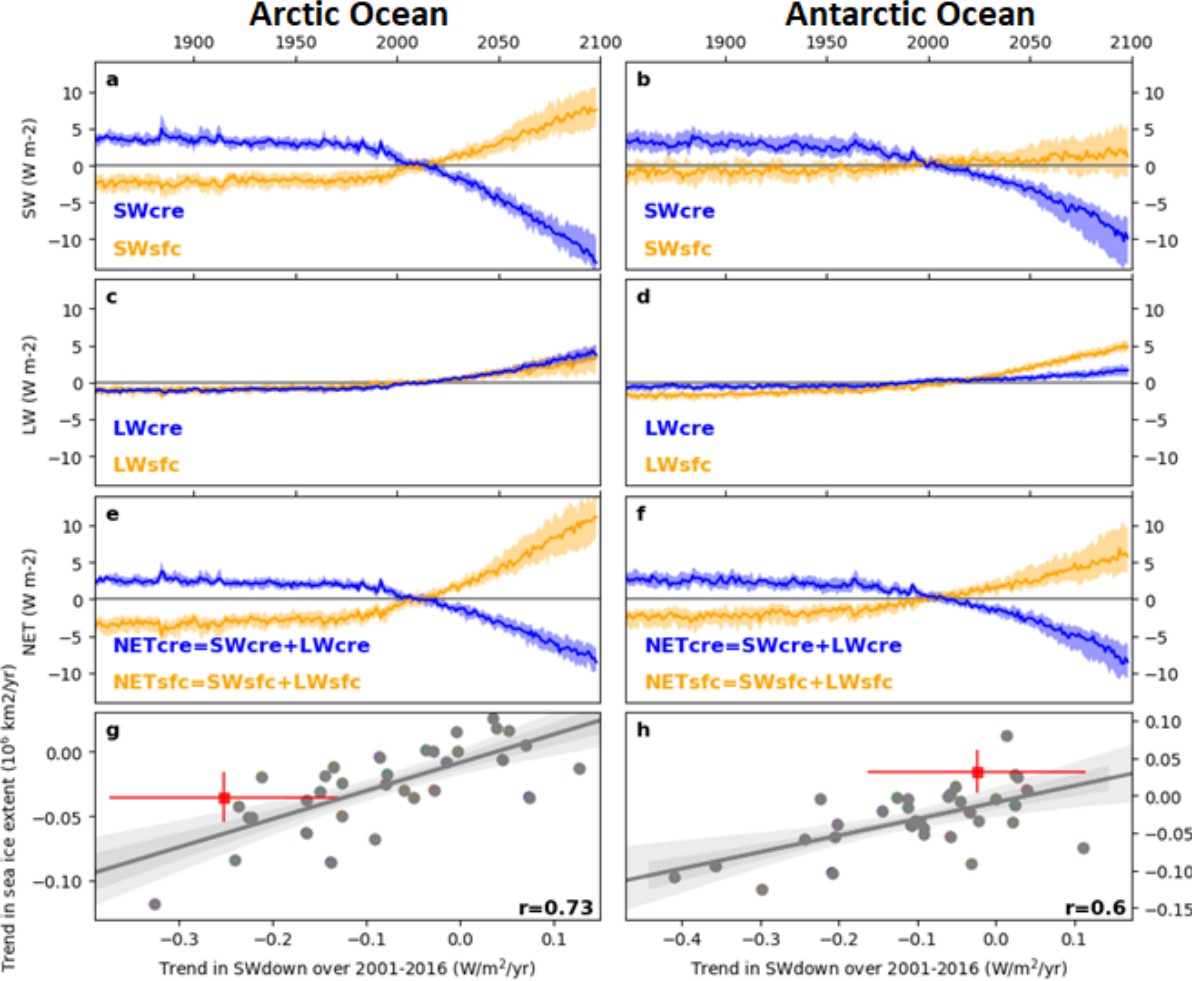

**Figure 9** Time series of the anomaly with respect to the whole period 1850-2100 of the radiative flux. Mean modeled SWcre, LWcre and NETcre (blue) and surface SWsfc, LWsfc and NETsfc (orange) anomalies over the 1850-2100 period under RCP8.5 scenario averaged over the Arctic Ocean. The solid line shows the median, where the envelope represents the 25 and 75 percentile of the 32 CMIP5 models. The linear regression (grey solid line and its 68% (dark grey envelope) and 95% (light grey envelope) confidence interval) between: the trend in SWdown and trend in sea ice extent (g and h); of the 32 CMIP5 climate models shown by grey dots over 2001-2016. The observed trends are shown by red colors where confidence interval refers to standard error of the trend.

**Acknowledgments:** The authors acknowledge the use of Clouds and the Earth's Radiant Energy System (CERES) satellite data version 4.0 from https://ceres.larc.nasa.gov/index.php, sea ice concentration data from National Snow and Ice Data Center (NSIDC) http://nsidc.org/data/G02202, as well as the modeling groups that contributed to the CMIP5 data archive at PCMDI https://cmip.llnl.gov/cmip5/.

**Author Contributions:** RA directed the study with contributions from all authors. RA performed the analysis. RA, PCT, AC and GD drafted the paper. All authors commented on the text.

**Competing interests:** The authors declare no competing financial interests.

**Additional information:** The programs used to generate all the results are made with Python. Analysis scripts are available by request to R. Alkama.

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
