# Peer review of "Clouds damp the radiative impacts of Polar sea ice loss 1 2 3 4 Authors: Ramdane Alkama1\*, Patrick C. Taylor2\*, Lorea Garcia-San Martin1, Herve Douville3, 5 Gregory Duveiller1, Giovanni Forzieri1, Didier Swingedouw4"

_The Cryosphere, 2019_

## Referee Comment (RC1) · Anonymous Referee #1 · 8 Jan 2020

The article presents a feedback atmospheric process following the decrease in sea ice concentration. The feedback begins with the change in sea ice concentration, followed by the surface energy balance change that changes cloud condition, then back to the surface energy balance. The feedback process presented in this paper roughly halves the direct consequence of the sea ice reduction, through cloud radiative effect. The article is an important contribution for evaluating the consequence of the on-going sea ice reduction, in a more realistic way than so far published works. For improving the realiability of presented numericals and also for easier readability by the workers in other fields however, minor alterations are suggested as listed below:

Scientific aspect: 1) The recgnition of clouds is a key point of this work. It is necessary to present how the CERES evaluation recognises the clouds. There are manuals stating this process, but a brief summary of the process in one paragraph will help readers. 2) Surface fluxes, whether through satellites or model computations, are subject to er-

[Figure]

rors that are often large. The quoted papers in the reference list do not satisfy this test. This reviewer recommends the authors to make a point-by-ponit comparison with the first-class ground observations. The sites, Ny-Alesund, Barrow, Alert and Resolute have long-standing observations of high quality irradiances for the Arctic. Similarly, Neumeyer, Syowa and South-Pole offers high quality irradiances with additional cloud information. The data are available at BSRN Centre at AWI, Bremerhafen.

Presentation and minor typological comments: P2, L 63 and elsewhere: It is necessary to provide the full names of ACRONYMs at their first appearences, e.g., CMIP on this page and P3 L 75 EBAF. P 3, Figure 1: To be consistent with the text, Swcre and Lwcre should read SWcre and LWcre. P4, L 98: The quoted publication, Kato et al. (2013) barely offers the information on the accuracy of irradiances, nor any of the authors are experienced with radiation science. P6, L 149: This sentence appears incomplete, or some words may have gone lost. P12, L223-224: This sentence is difficult to understand. P14, L 310: "half if induced by" may read "half is induced by". P15, L 317: "should aim to reduce" may read better when "should aim at reducing". P18, L 390: Too many authors presented. This paper was written by four authors only.

These are, however a minor comments, and this reviewer hopes that the authors will work for the quickest publication of this interesting work.

---

## Referee Comment (RC2) · Anonymous Referee #2 · 8 Mar 2020

The study investigates the correlation and covariation between cloud radiative effect (CRE) and sea ice in the Arctic and Antarctic using satellite and climate model data. It is found that clouds play a significant role in damping the net change in radiation absorbed at the surface as a result of sea ice changes.

It is an interesting study, but I have issues with the interpretation and the manuscript is not particularly well written. There are language issues that need to be worked on and the methodology and logical steps need to be explained better. The results are interesting and potentially of some importance for our interpretation of the ensemble spread in polar climate responses in the CMIP-archives. While I do not think many new analyses are needed, I do fear that some of the interpretations are to bold and need to be moderated. Therefore, I cannot advise to accept without major revisions of the manuscript.

Major points

You write in the abstract that "years with less sea ice and a larger net surface radiative flux are also those that show an increase in sunlight reflected back to space by clouds." I am not convinced that this is, in fact, what you find. I would rather say that they are the years with a larger CRE. This is not the same, since as you point out when discussing mechanism (I) in L157 onwards: Even if cloud properties are held constant, the CRE can change due to the changes in clear-sky radiation induced by changes in sea ice decline and surface albedo.

When surface albedo is lowered, more of the sunlight passing through the atmosphere is absorbed at the surface resulting in greater SW_total. But SW_clear increases even more since the lower albedo allows a larger fraction of the extra downwelling SW at the surface to be absorbed. This means that the quantity SW_cre = SW_total – SW_clear is decreased even in the absence of cloud changes – a purely surface-related effect.

I believe the above quoted statement ignores this; a point which is reflected in the next sentence: "An increase in absorbed solar radiation when sea ice retreats (surface albedo change) explains $66 \pm 2\%$ of the observed signal". As I understand your analyses, these 66% are exactly this surface-only effect. So the "observed signal" referred to in this sentence is the signal in CRE and not in "sunlight reflected back to space by clouds" as the previous sentence suggests.

I believe this is not just a matter of wording. I think it really is an important part of how the results are interpreted and served to the reader. I will therefore give more examples where this distinction is not made clearly enough throughout the manuscript:

L187: "We estimate that the cloud changes in the Antarctic system are damping by 56% . . .". Here, "cloud changes" should be replaced by "the existence of clouds and the changes therein" or something to that effect, since as I understand it, the existence accounts for two thirds of the effect and the changes for only one third. Right?

[Figure]

L223: "polar sea ice and cloud covarying in a way that substantially reduces the overall impact of the sea ice loss". Again, as far as I can see, only a third of the effect is due to the covariance. Two thirds is just due to clouds being present.

L245:" We argue that the strong increase of SWcre under decreased sea ice observed during summer is induced by larger values of cloud optical depth (Fig. 7a)". Again, what about process (I)?

L309 (conclusion): "Satellite data indicates that the increased cloud cover/thickness correlates with sea ice melting, reducing by half the potential increase of net radiation at the surface". I think your results show that, only 33% of the by-half-reduction is due to changed clouds, while the remaining 66% is due to the mere presence of clouds.

Minor points

L37 Introduction: You should look into the results of Qu and Hall (2006, JClimate) who in their figure 6a illustrate that across a climate model ensemble, planetary albedo variations resulting from surface albedo variations are muted by half. While this study focused on terrestrial albedo variations due to snow changes, the point is the same: The mere existence of clouds damp the TOA effect of surface albedo variations. This is similar enough to your findings that they ought to be discussed in the context of your results. Either in the intro, discussion or conclusions.

Figure 1: In the equations below panel b, I believe you have ordered the terms on the RHS wrong: Shouldn't it be SWtotal-SWclear, and LWtotal-LWclear?

L72 "Methods and data": As this section is currently, you talk a lot about the data but not really about the methods you will use. Then you go directly to the "Results and discussions" section which is difficult to read because the entire methodology is left in the supplement. I believe your statistical methods and your plots are so non-standard, and not least completely central to your analyses and conclusions, that they should be lifted from the supplement and into the "Methods and data" section.
L78-80: You need to explain the sign convention of the fluxes explicitly. I assume all fluxes are taken positive downwards, but it does not say so anywhere and while Figure 1 does say, for instance, LWclear at the end of a red arrow, this does not explain the sign convention. If anything, it is a bit confusing since this makes it look as if the LW's are taken positive upward.

Figure 3: How are the models ordered? Not alphabetically, it seems.

L157: Mechanism (I) is really important to the paper (as discussed above). Given this major importance, the explanation of the mechanism is not clear enough, so that all readers understand it. It becomes too easy for the reader to misunderstand it and think that is actually has something to do with the clouds when it really is a surface-only phenomenon. Please restructure the paragraph explaining mechanisms I and II such that you give yourself room enough to do it properly. Also, you have made the nice schematic in Figure 1. Use this and point to it in your explanations.

L164-166: This sentence assumes the reader is familiar with how to read Fig 4, something we are not until we have read the supplement. Lifting this into Section 2 would help a lot – but at least be clear and tell the reader that the supplement is, in fact, a prerequisite for understanding the entire paper.

L183-184: Units on the equations?

L190-191: Here you just add the errors but I am unsure whether you shouldn't, in fact, be adding them in quadrature. This, of course, depends on whether you believe the errors to be correlated or not. Please consider this carefully.

Figure 7: Is the data (or the methodology behind it) in this figure taken directly from Taylor et al? If so, please say so. Otherwise, the reader searches this paper for details in vain.

L294/Figure 8gh: You do not discuss the red cross in Figure 8 gh. Why then show it? If you have a point with this information, you need to discuss it in the text. Otherwise,

remove it from the figure.

L296: "This analysis suggests that the models showing a larger trend in cloud cover also show larger decreases in sea-ice extent and clearly demonstrate the strong coupling of these two variables.": Yes, but do you propose anything in terms of causality between the two? If not, you should be clear about this. Otherwise, the reader may try and read between the lines here.

L316: "that show smaller trends in surface". Shouldn't this be larger trends? That is, at least, what I get out of L295.

Figure 8: The time series are anomalies, but with respect to which period?

Suppl. L54: optical depth.

Suppl. L73: What are M and N? You do not seem to say so.

Suppl L78: $A_i$ is the total area of all grid cells with a particular SIC change, right? Please explain this better.

Suppl L90: What is $SX_p$?

Language:

There are many examples of language that is not quite at an acceptable level. I cannot list them all, but I urge you to have a native English-speaker go carefully through the manuscript. Examples are

L 20: clouds ->cloud

L21: responding

L23: "manner of the" sounds weird. Please rephrase.

L45: determines

L58: Alternatively -> On the other hand

---

## Author Response (AR1)

Please find below the referees comments (in black) and our answers (in blue).

**Anonymous Referee #1**

The article presents a feedback atmospheric process following the decrease in sea ice concentration. The feedback begins with the change in sea ice concentration, followed by the surface energy balance change that changes cloud condition, then back to the surface energy balance. The feedback process presented in this paper roughly halves the direct consequence of the sea ice reduction, through cloud radiative effect. The article is an important contribution for evaluating the consequence of the on-going sea ice reduction, in a more realistic way than so far published works.

We thank the reviewer for her (his) positive comments.

For improving the realiability of presented numericals and also for easier readability by the workers in other fields however, minor alterations are suggested as listed below:
Scientific aspect:

1) The recgnition of clouds is a key point of this work. It is necessary to present how the CERES evaluation recognises the clouds. There are manuals stating this process, but a brief summary of the process in one paragraph will help readers.

Additional discussion and references to the cloud retrieval techniques are provided in Section 2.1.

2) Surface fluxes, whether through satellites or model computations, are subject to errors that are often large. The quoted papers in the reference list do not satisfy this test. This reviewer recommends the authors to make a point-by-ponit comparison with the first-class ground observations. The sites, Ny-Alesund, Barrow, Alert and Resolute have long-standing observations of high quality irradiances for the Arctic. Similarly, Neumeyer, Syowa and South-Pole offers high quality irradiances with additional cloud information. The data are available at BSRN Centre at AWI, Bremerhafen.

Thank you for this comment. We also agree that the determination of radiative surface fluxes using satellite data is a challenge prospect. The CERES science team has spent much of the last 20 years analysing and refining these data. The requested comparisons have been undertaken and published by the CERES Science Team (e.g., Kato et al. 2018). Kato et al. (2018) compared the CERES surface EBAF Ed 4 monthly mean surface radiative fluxes with 46 buoys and 36 land sites, including the high-quality sites in the Arctic (e.g, Ny-Alesund, Barrow, Alert, and Resolute). The uncertainty estimates for individual surface radiative flux terms in the Arctic range from 12-16 W m$^{-2}$ (1$\sigma$) at the monthly mean 1$^o$x$^o$1 gridded scale. Moreover, previous studies have stated that the CERES SFC EBAF fluxes are as a key benchmark for evaluating the Arctic surface radiation budget (Boeke et al. 2016; Christensen et al. 2016; Duncan et al. 2020). This discussion is now included in the text.

References:
Kato, S. and coauthors, 2018: Surface Irradiances of Edition 4.0 Clouds and the Earth's Radiant Energy System (CERES) Energy Balanced and Filled (EBAF) Data Product. *J. Climate*, **31**, 4501-4527, doi: 10.1175/JCLI-D-17-0523.1.

Boeke, R. C. and P. C. Taylor, 2016: Evaluation of the Arctic surface radiation budget in CMIP5 models. *J. Geophys. Res.,* **121**, 8525-8548, doi: 10.1002/2016JD025099.

Christensen, M., A. Behrangi, T. L'Ecuyer, N. Wood, M. Lebsock, and G. Stephens (2016), Arctic observation and reanalysis integrated system: A new data product for validation and climate study, Bull. Am. Meteorol. Soc., doi:10.1175/BAMS-D-14-00273.1.

Duncan, B. N., Ott, L. E., Abshire, J. B., Brucker, L., Carroll, M. L., Carton, J. and coauthors, 2020: Space-based observations for understanding changes in the arctic-boreal zone. *Reviews of Geophysics*, 58, e2019RG000652. https://doi.org/10.1029/2019RG000652

Presentation and minor typological comments: P2, L 63 and elsewhere: It is necessary to provide the full names of ACRONYMs at their first appearences, e.g., CMIP on this page and P3 L 75 EBAF. P 3,
OK, done see lines 72-74 and 85.

Figure 1: To be consistent with the text, Swcre and Lwcre should read SWcre and LWcre.
OK, done see Figure 1.

P4, L 98: The quoted publication, Kato et al. (2013) barely offers the information on the accuracy of irradiances, nor any of the authors are experienced with radiation science.

Additional references describing and analysing the CERES SFC EBAF data have been added to the manuscript along with a more detailed description of the surface radiative flux uncertainty, also see previous response. The reviewer should also know that the author list includes a member of the CERES Science Team experienced with radiation science. See section 2.1.

P6, L 149: This sentence appears incomplete, or some words may have gone lost.
P12, L223-224: This sentence is difficult to understand.
P14, L 310: "half if induced by" may read "half is induced by".
P15, L 317: "should aim to reduce" may read better when "should aim at reducing".
P18, L 390: Too many authors presented. This paper was written by four authors only.
Ok, done. Thanks

These are, however a minor comments, and this reviewer hopes that the authors will work for the quickest publication of this interesting work.
We thank the reviewer for his constructive comments that allows us to improve he manuscript.

Please find below the referee comments (in black) and our answers (in blue).

**Anonymous Referee #2**

The study investigates the correlation and covariation between cloud radiative effect (CRE) and sea ice in the Arctic and Antarctic using satellite and climate model data. It is found that clouds play a significant role in damping the net change in radiation absorbed at the surface as a result of sea ice changes.
It is an interesting study, but I have issues with the interpretation and the manuscript is not particularly well written. There are language issues that need to be worked on and the methodology and logical steps need to be explained better. The results are interesting and potentially of some importance for our interpretation of the ensemble spread in polar climate responses in the CMIP-archives. While I do not think many new analyses are needed, I do fear that some of the interpretations are to bold and need to be moderated. Therefore, I cannot advise to accept without major revisions of the manuscript.

Major points
You write in the abstract that "years with less sea ice and a larger net surface radiative flux are also those that show an increase in sunlight reflected back to space by clouds." I am not convinced that this is, in fact, what you find. I would rather say that they are the years with a larger CRE. This is not the same, since as you point out when discussing mechanism (I) in L157 onwards: Even if cloud properties are held constant, the CRE can change due to the changes in clear-sky radiation induced by changes in sea ice decline and surface albedo. When surface albedo is lowered, more of the sunlight passing through the atmosphere is absorbed at the surface resulting in greater SW_total. But SW_clear increases even more since the lower albedo allows a larger fraction of the extra downwelling SW at the surface to be absorbed. This means that the quantity SW_cre = SW_total – SW_clear is decreased even in the absence of cloud changes – a purely surface-related effect.
I believe the above quoted statement ignores this; a point which is reflected in the next sentence: "An increase in absorbed solar radiation when sea ice retreats (surface albedo change) explains $66 \pm 2\%$ of the observed signal". As I understand your analyses, these 66% are exactly this surface-only effect. So the "observed signal" referred to in this sentence is the signal in CRE and not in "sunlight reflected back to space by clouds" as the previous sentence suggests.

We strongly agree with the reviewer. As noted by the reviewer, we state in the manuscript that surface albedo changes can drive substantial change in the cloud radiative effect and that such a change can occur in the absence of a change in cloud properties. Through the initial writing of the manuscript we went great lengths to try to unsure that this interpretation was clear. However, it is clear that we have missed a few statements. This point is extremely important, as it can change our interpretation, yet nuanced at the same time. We have gone through the manuscript to correct all instances of this. As she (he) suggested, we replaced "an increase in sunlight reflected back to space by clouds" by "larger cloud radiative effect" see line 25.

I believe this is not just a matter of wording. I think it really is an important part of how the results are interpreted and served to the reader. I will therefore give more examples where this distinction is not made clearly enough throughout the manuscript:
L187: "We estimate that the cloud changes in the Antarctic system are damping by 56% . . .".
Here, "cloud changes" should be replaced by "the existence of clouds and the changes therein" or something to that effect, since as I understand it, the existence accounts for two thirds of the effect and the changes for only one third. Right?

We agree that this is an important point and does change the interpretation of the results as well as the implications of the manuscript. We have tried to make this point clear by rewriting the abstract as to highlight this point.

From lines 28-32: "Thus, present-day cloud properties significantly reduce the net radiative impact of sea ice loss on the Arctic and Antarctic surface radiation budgets. As a result, climate models must accurately represent present-day polar cloud properties in order to capture the surface radiation budget impact of polar sea ice loss and thus the surface albedo feedback."

In addition, the text in lines 263-265 specifically calls out "Thus, the observed negative correlation between SWcre and SWsfc over the polar seas results from the larger effects of process (I) than (II)." Also, as suggested by the reviewer "cloud changes" is replaced by "the existence of clouds and their property variations" (see line 277).

L223: "polar sea ice and cloud covarying in a way that substantially reduces the overall impact of the sea ice loss". Again, as far as I can see, only a third of the effect is due to the covariance. Two thirds is just due to clouds being present.

Ok, done. This sentence is replaced by "the results suggest clouds substantially reduce the impact of sea ice loss on the surface radiation budget and thus the observed sea the sea ice albedo feedback" see lines 314-315.

L245:" We argue that the strong increase of SWcre under decreased sea ice observed during summer is induced by larger values of cloud optical depth (Fig. 7a)". Again, what about process (I)?

Ok, done "SWcre" is replaced by "SWcreCloud", see line 339.

L309 (conclusion): "Satellite data indicates that the increased cloud cover/thickness correlates with sea ice melting, reducing by half the potential increase of net radiation at the surface". I think your results show that, only 33% of the by-half-reduction is due to changed clouds, while the remaining 66% is due to the mere presence of clouds.

In the new version of the manuscript, the words "Cloud cover/thickness" are replaced by "cloud radiative effect". See line 408.

Minor points

L37 Introduction: You should look into the results of Qu and Hall (2006, JClimate) who in their figure 6a illustrate that across a climate model ensemble, planetary albedo variations resulting from surface albedo variations are muted by half. While this study focused on terrestrial albedo variations due to snow changes, the point is the same: The mere existence of clouds damp the TOA effect of surface albedo variations. This is similar enough to your findings that they ought to be discussed in the context of your results. Either in the intro, discussion or conclusions.

This reference is included in the in the new version of the manuscript in the discussion when discussing the TOA dampening effect (see lines 317-319). During the review of this manuscript, we became aware of a recent paper by Sledd and L'Ecuyer (2019). This paper uses reanalysis output to quantify the "masking" or damping effect of clouds on the radiative effect of surface albedo variability. There result corroborates our result arguing that clouds damp the effect of surface albedo variability on top-of-atmosphere albedo (reflected shortwave flux) by half. This reference has also been added to the manuscript (see lines 319-321).

Reference:

Sledd and L'Ecuyer, 2019: How Much Do Clouds Mask the Impacts of Arctic Sea Ice and Snow Cover Variations? Different Perspectives from Observations and Reanalyses. https://www.mdpi.com/2073-4433/10/1/12/htm#B28-atmosphere-10-00012

Figure 1: In the equations below panel b, I believe you have ordered the terms on the RHS wrong: Shouldn't it be SWtotal-SWclear, and LWtotal-LWclear?

Ok, done see new figure 1.

L72 "Methods and data": As this section is currently, you talk a lot about the data but not really about the methods you will use. Then you go directly to the "Results and discussions" section which is difficult to read because the entire methodology is left in the supplement. I believe your statistical methods and your plots are so non-standard, and not least completely central to your analyses and conclusions, that they should be lifted from the supplement and into the "Methods and data" section.
Ok, methods are moved from supplementary to Methods and data (see section 2.7 and 2.8).

L78-80: You need to explain the sign convention of the fluxes explicitly. I assume all fluxes are taken positive downwards, but it does not say so anywhere and while Figure 1 does say, for instance, LWclear at the end of a red arrow, this does not explain the sign convention. If anything, it is a bit confusing since this makes it look as if the LW's are taken positive upward.
Ok, done (see lines 81 and 106-107).

Figure 3: How are the models ordered? Not alphabetically, it seems.
The models are ordered alphabetically in the new version of the manuscript (see actual figure 4).

L157: Mechanism (I) is really important to the paper (as discussed above). Given this major importance, the explanation of the mechanism is not clear enough, so that all readers understand it. It becomes too easy for the reader to misunderstand it and think that is actually has something to do with the clouds when it really is a surface-only phenomenon. Please restructure the paragraph explaining mechanisms I and II such that you give yourself room enough to do it properly. Also, you have made the nice schematic in Figure 1. Use this and point to it in your explanations.
Ok done, see lines 242-252.

L164-166: This sentence assumes the reader is familiar with how to read Fig 4, something we are not until we have read the supplement. Lifting this into Section 2 would help a lot – but at least be clear and tell the reader that the supplement is, in fact, a prerequisite for understanding the entire paper.
In the new version of the manuscript we moved the method section from the supplement to the Method and data section 2.7. We also referred to this section in lines 255, 293, 305 and 363.

L183-184: Units on the equations?
Ok done, see lines 271-272.

L190-191: Here you just add the errors but I am unsure whether you shouldn't, in fact, be adding them in quadrature. This, of course, depends on whether you believe the errors to be correlated or not. Please consider this carefully.
Ok, done, see line 280.

Figure 7: Is the data (or the methodology behind it) in this figure taken directly from Taylor et al? If so, please say so. Otherwise, the reader searches this paper for details in vain.
Ok done see lines 96-100 and 363, data from CERES and method as described in section 2.7.

L294/Figure 8gh: You do not discuss the red cross in Figure 8 gh. Why then show it? If you have a point with this information, you need to discuss it in the text. Otherwise, remove it from the figure.
New sentence "We also note that from the 32 models tested, only few show consistent trends in both SWdown and SIC over 2001-2016 (Figure 9gh)" is included in the new version of the manuscript (see lines 392-393).

L296: "This analysis suggests that the models showing a larger trend in cloud cover also show larger decreases in sea-ice extent and clearly demonstrate the strong coupling of these two variables.": Yes, but do you propose anything in terms of causality between the two? If not, you should be clear about this. Otherwise, the reader may try and read between the lines here.
Ok, the sentence "However, the direction of causality between the two variables is unclear" is included in the new version of the manuscript. See lines 391-392.

L316: "that show smaller trends in surface". Shouldn't this be larger trends? That is, at least, what I get out of L295.
We agree, larger is the correct word. Thus, smaller is replaced by larger in the new version of the manuscript. See line 414.

Figure 8: The time series are anomalies, but with respect to which period?
The anomalies in respect to the whole period. This is clearly stated in the new version of the manuscript. See line 434.

Suppl. L54: optical depth.
This is moved to the main text, see line 195.

Suppl. L73: What are M and N? You do not seem to say so.
Ok done see lines 149 and 155.

Suppl L78: A_i is the total area of all grid cells with a particular SIC change, right? Please explain this better.
To avoid a misunderstanding between $i$ of year $yi$ with the one used in equations, we replaced i in the equations by j.
$\sum_{j=1}^{N} A_j$ is the total area of all grid cells with a particular SIC change. In case of 60% SIC change with an increment of 10%, for example, $\sum_{j=1}^{N} A_j = A_1 + A_2 + A_3 + A_4 + A_5$

$A_1$ is the total area of all grid cells with SIC change from 60% to 0% in two consecutive years.

$A_2$ is the total area of all grid cells with SIC change from 70% to 10% .

$A_3$ is the total area of all grid cells with SIC change from 80% to 20%.

$A_4$ is the total area of all grid cells with SIC change from 90% to 30%.

$A_1$ is the total area of all grid cells with SIC change from 100% to 40%.

Actual figure 2 and line 158 explains this in better way than the old version.

Suppl L90: What is SX_p?
As mentioned in previous lines 85-89 actual lines 169-174 "S" is the slope while Xp is the relative change in sea ice concentration.

Language:
There are many examples of language that is not quite at an acceptable level. I cannot list them all, but I urge you to have a native English-speaker go carefully through the manuscript. Examples are
L 20: clouds ->cloud
L21: responding
L23: "manner of the" sounds weird. Please rephrase.
L45: determines
L58: Alternatively -> On the other hand

Ok, done. Thanks. Experienced scholarly writer P. C. Taylor who is *native English-speakers* have edited this new version of the manuscript.

We thank the reviewer for the constructive comments that helped us to improve the manuscript.

[revised manuscript text omitted]

correlation between inter-annual [garbled] NET cre [garbled] SW&LW [garbled] over much of the [garbled]
CERES TOA flux data from 2001-2016. However, our analysis reveals the opposite pattern over
the polar seas (defined in section 2.5) where the correlation is negative over the Antarctic and
partly negative over the Arctic (Bering Strait, Hudson Bay, Barents Sea and the Canadian
Arctic) [garbled overlapping text] SW and LW [garbled] cre SW
(Fig. 2cd3cd) shows a similar patternspattern of correlation as beforethe NET cre (Fig. 2ab3ab) but with a stronger
magnitude, while LW cre generally experienceshows the opposite correlations (Fig. 2ef3ef). This
suggests that SW radiation fluxes the factors influencing SW cre are responsible for the sharp
contrast betweenin the correlation found in the polar regions and the rest of the world.. Indeed,
SW sfc and SW cre (Fig. 2gh3gh) show the sharpest and most significant contrast between the polar
regions and the rest of the world (Fig. S2 is similar to Fig 2. 3 but only significant
correlationcorrelations at the 95% confidence level are reported in blue and red colors). On
averageOverall, climate models are able to reproduce the spatial pattern of the observed SW
correlation, but also show a large inter-model spread concerningin the spatial extent of the
phenomena (Fig. 34 and S3). On the other hand, several models completely fail at reproducing this
fundamentalto reproduce the correlation. Indeed, ACCESS1-3, MIROC5, CanESM2 and CSIRO-
Mk3-6-0 models showsshow negative correlation over Antarctic continent in contrast to observed
positive correlation. Also, someSome models, like IPSL-CM5B-LR, GISS-E2-R and bcc-csm1-1
completely, fail to reproduce the observed negative correlation over the Southern Ocean. This
suggests that these models madecontain misrepresentations of the relationships SW cre and NET sfc
likely resulting from errors in simulating the relationships between sea ice extend, surface albedo,
cloud cover/thickness, and/or their relationships between influence on surface radiative flux and
cloud properties, whichfluxes that could severely impact their projections. Moreover, Fig. 4
demonstrates that simple correlations between NET sfc and the individual radiation budget terms
represents a powerful metric for climate model evaluation allows for a quick check for realistic
surface radiation budget variability in polar regions.

[Figure]

**Figure** 3 Correlation between TOA CRE and surface radiation budget terms over 2001-2016 from CERES measurements for the Northern Hemisphere (aceg) and Southern Hemisphere (bdfh) polar sea. Positive correlations  red color indicate that  years with less NETsfc coincide with  years where NETcre has a stronger cooling effect and *vice versa*.

[Figure]

**Figure** 4 Correlation between SWcre and  SWsfc shown by 32
CMIP5 earth system models and  CERES between 2001 and 2016 over
the Southern Hemisphere.

**3.2 Effects of sea ice concentration change**

We illustrate that the apparent contradiction over the polar seas between NET cre and NET sfc found in Fig 3ab is caused by the factors contributing to the SW fluxes. This can be explained by: (I) SW cre can change even if cloud properties are held constant due to the changes in clear-sky radiation induced by changes in sea ice (and surface albedo). When surface albedo is reduced, the surface absorbs more sunlight at the surface resulting in a greater SW total. At the same time, SW clear increases since the lower albedo allows a larger fraction of the extra downwelling SW at the surface to be absorbed (see Fig. 1). Therefore, SW cre becomes more negative even in the absence of cloud changes (a purely surface-related effect); (II) On the other hand, the relationship between cloud cover/thickness and sea ice could lead to cloudier Polar seas under melting sea ice (Abe et al., 2016; Liu et al., 2012) such that the SW cre decreases (increasing the amount of SW reflected back to space by clouds, see Fig. 1), thus the cloud cooling effect is enhanced concurrently with melting sea ice (a purely cloud-related effect). Both of these factors occur simultaneously.

Over the Antarctic seas, analysis of the year-to-year changes in SW down stratified in 2% SIC bins retrieved from satellite microwave radiometer measurements (see section 2.7) shows an increase in SW down with increased SIC and *vice-versa* (Fig. 5a). This suggests that years with higher SIC also have fewer and/or thinner clouds (Liu et al., 2012) (Fig. 6), larger SW down, and also larger upward SW radiation (SW up) (Fig. 5b), due to higher surface albedo (Fig. S4). Consequently, these years show a more negative SW sfc (Fig. 5c) and thus are characterized by stronger surface cooling. Furthermore, fewer clouds implies a reduction of the cloud cooling effect (less negative SW cre) as described above in process (II), this accounts for $34 \pm 1\%$ (Fig. 7d) of the total change in SW cre, and as described in process (I) the increase in the surface albedo also makes SW cre less negative and explains $66 \pm 2\%$ of the observed change (Fig. 7d). Thus, the observed negative correlation between SW cre and SW sfc over the polar seas results from the larger effects of process (I) than (II). Similar results are found over the Arctic Ocean with slightly different sensitivity (Fig. S5, S6). This difference is tied to differences in sun angle/available sunlight, as Antarctic sea ice is concentrated at lower latitudes than Arctic sea ice.

Using the regression relationships derived from our composite analysis, we estimate the magnitude of the cloud effect. For the Antarctic system, we use the numbers found in Figure 5e, where we find the annual mean relationship between NET sfc (in $W/m^2$) and SIC (fraction between 0 and 1), and NET cre (in $W/m^2$) and SIC (fraction between 0 and 1).

$\Delta$NET sfc=(-36.61±0.72)$\Delta$SIC (18)

$\Delta$NET cre=(47.03±1.01)$\Delta$SIC (19)

In case of When excluding the CRE, the ΔNET sfc would be equal to (-36.61-47.03) ΔSIC =-83.64 Δ
[revised manuscript text omitted]

---

## Author Response (AR2)

Please find below the referee 2 comments (in black) and our answers (in blue).

**Anonymous Referee #2**

L183+196: This equality is only approximate (to 1. order), right? Perhaps use a \simeq.

Ok, done. See lines 183-196

L242-3: "changes in clear-sky radiation changes". One too many "changes".

Ok, done. The second "changes" is deleted from the text. See line 242-243.

L261+263: You give the numbers 34% and 66% and the only reference you give is Fig 7d. I guess I am supposed to be able to derive the 34 and 66 from the numbers of the fits printed on Fig 7d, and here I have two issues: i) the numbers are extremely small and difficult to read and ii) please help the reader how to derive the 34 and 66. These are central numbers from the study and you do not want to leave any doubt about them with the reader.

Ok, done. $34\% = (19.42 * 100)/56.59$ and $66\% = (37.1742 * 100)/56.59$. The numbers came from Fig 7d bottom. See lines 261-264.

L315. Strike "sea the"

Ok, removed.

L336-337: "strong increases … are limited". Sounds odd. Either "strong increases … are excluded/absent/etc" or just "increases… are limited".

Ok, done. We opted for the second suggestion such as "increases… are limited". See line 337

Fig 8: In many panels, the legends cover the data, lines and boxes. If you want the reader to see the data, please change this.

Ok, done. See new figure 8.

Sect 3.4: This whole section could use another round on the language. It appears to not have been gone over as thoroughly as the rest of the paper.

Ok, done. Experienced scholarly writer P. C. Taylor who is *native English-speakers* have edited this section in the new version of the manuscript.

L378: "over THE Arctic Ocean"

Ok, done.

L378: "Arctic Ocean", in the figure (Fig 9) you call it "Arctic sea" (which also sounds odd). Please choose one and stick with it.

Ok, Arctic Sea is replaced by Arctic Ocean in the new version of the manuscript. See figure 9

L379: "dampening", in other places you say "damping"

Ok, "dampening" is replaced by "damping" in the new version of the manuscript.

L381: "causes"

Ok, done. See line 382.

L382: "Fig 9gh": Are panels g and h really the right reference for this sentence?

(Fig. 9ef) is the correct reference. We corrected this in line 383

L393: "consistent": Consistent with what? With the observed trends? If so, say so.
Yes, consistent with observed trends. This is clearly mentioned in line 394 of the new version of the manuscript.

L404: "sea-ice" In the rest of the paper you write "sea ice". Choose one.

We choose sea ice in the new version of the manuscript.

L406: "between two consecutive years, the cloud radiative...".
Ok, done. See line 407.

L422: Why is the Solomon reference written differently from the others?

Ok, we corrected this in line 423 of the current version of the manuscript.

L426: "At the very least,". I think you are being too modest here. I would suggest something like "On a practical level,".

Ok, replaced. See line 426.

L434: "with respect"

Ok, done. See line 435.

L436: "RCP8.5" – that is also how you write it in L371.

Ok, done. See line 437.

We thank the reviewer her (his) edits and comments that helped us to improve the manuscript.

[revised manuscript text omitted]